# Hybrid Coupler Used as Tunable Phase Shifter Based on Varactor Diodes

**DOI:** 10.3390/mi15070838

**Published:** 2024-06-28

**Authors:** Taleb Mohamed Benaouf, Abdelaziz Hamdoun, Mohamed Himdi, Olivier Lafond, Hassan Ammor

**Affiliations:** 1ERSC Team, Mohammadia School of Engineers, Mohammed V University of Rabat, Rabat 10100, Morocco; ammor@emi.ac.ma; 2XLIM UMR CNRS 7252, University of Poitiers, 86073 Angoulême, France; abdelaziz.hamdoun@univ-poitiers.fr; 3Institut d’Electronique et des Technologies du numeRique (IETR), UMR CNRS 6164, Université de Rennes, 35000 Rennes, France; mohamed.himdi@univ-rennes1.fr (M.H.); olivier.lafond@univ-rennes1.fr (O.L.)

**Keywords:** reconfigurable coupler, coupler hybrid 3 dB, phase shifter

## Abstract

This paper describes the design of a hybrid coupler with a continuously variable output phase difference. This is achieved by using reconfigurable transmission lines with electrically tunable lengths controlled by two biasing voltages through varactor diodes placed across the coupler branches. The design of the coupler is based on the quadrature hybrid structure for the case where the output phase difference is 90° and on the asymmetric structure for the other cases. The proposed coupler can achieve a tunable output phase difference from 52° to 128°, while keeping a coupling coefficient of −3 dB (± 0.5 dB) over the entire desired frequency band. To validate the simulated results, a prototype working at 3.5 GHz was fabricated and tested. The measurement results show good correspondence with the simulation results, especially when the output phase difference is 90°, while a phase mismatch of less than 7° was observed for the other cases. The presented coupler would be a great asset for antenna feeding arrays, especially the Butler matrix.

## 1. Introduction

Hybrid couplers are among the most important passive microwave devices. They are widely used in various RF and microwave systems. They operate by dividing or combining power with a constant phase shift [1]. These features are particularly useful in various applications such as antenna feed arrays, especially for one of the best known, the Butler matrix.

Since the Butler matrix is highly dependent on couplers, it is clear that the development of couplers will improve the capabilities of the matrix.

Several improvements have been made to hybrid couplers, e.g., couplers with a modified branch line to have a wide bandwidth [2], couplers with a reduced size [3,4] or reduced levels of sidelobes [5], couplers based on the fractal concept of composite right/left handed transmission line (CRLH TL) to reduce the size of the Butler matrix [6].

Alternative studies have concentrated on couplers that can be tuned in terms of frequency [7,8,9]. One proposal, as detailed in [7], involves using varactor diodes on the patch coupler’s patterned ground plane (PGP) to obtain frequency adjustability. A separate technique, highlighted in [8], deploys micro-electromechanical systems (MEMS) capacitors on the branch-line coupler to accomplish the same goal. To ensure a wider operating bandwidth, the vertically installed planar (VIP) structure was put into use [9].

Alongside frequency adjustability, having the ability to manage the signal amplitudes in wireless communication systems is also important [10]. As a result, couplers featuring adaptable coupling coefficients have been introduced. For example, a varactor-loaded PGP setup can be employed to attain tunable coupling coefficients [11]. In [12], a pair of varactors were stationed at the center of the altered coupled lines to broaden the tunable coupling coefficient range.

Moreover, the capacity to modify a coupler’s phase characteristics is a key to a multitude of applications, including the emerging beamforming system [13]. However, conventional couplers are limited to providing standard phase differences, such as 90° for quadrature couplers and 0° or 180° for rat-race couplers [14]. In order to simplify the network structure, some authors propose reducing the phase difference of the coupler outputs to 45° instead of 90° [15,16]. In addressing this issue, various coupler configurations have been proposed to offer nonstandard phase characteristics [17,18,19]. However, these configurations can only provide a fixed phase difference, limiting their flexibility. In [20], the researcher introduced hybrid couplers with electrically switchable phase differences based on six PIN diodes, which require dual sections and two distinct coupler structures to achieve the desired phase difference. Additionally, in [21] and [22], the authors propose couplers with tunable phase differences, but the first one, a dual-section structure is required to obtain a range from 60° to 120°, thus making the design more complex and increasing line losses. The second one offers a tunable coupler from 45° to 135° but at the cost of very high insertion losses. Moreover, both of these designs operate at very low frequencies. Clearly, a single, simple structure for a hybrid coupler with a phase difference that can be continuously adjusted as required would be of real interest. In this work, we propose a hybrid coupler with a simple planar structure, low cost, and ease of fabrication, with a continuously tunable phase difference ranging from 52° to 128° (meaning 90° ± 38°). This was obtained throughout various electrical lengths via different capacitive loadings from varactor diodes. 

The paper is organized into three sections. Section 1 presents a comprehensive analysis of the circuit configuration with theoretical insights. In Section 2, the proposed coupler is designed, fabricated, and tested for validation purposes. The final section, Section 3, summarizes the findings and presents the conclusion drawn from the analysis and measurements performed in the previous sections.

## 2. Design and Analysis

The design theory of the proposed coupler is achieved in two steps:Analysis and calculation of the new electrical lengths θ and the new impedance Z, obtained after adding the two variable capacitors, as illustrated in Figure 1b.Even–odd mode analysis to obtain the parameters of the coupler taking into account θ and Z.

### 2.1. Analysis and Calculation of θ and Z

To determine the electrical length θ and the corresponding impedance *Z*, we reuse the equations obtained in [23] while analyzing the equivalent Pi network circuit of a transmission line, as shown in Figure 1a:(1)$$\frac{C_{0}}{2}=\frac{\tan\left(\frac{\theta_{2}}{2}\right)}{Z_{2}\omega }$$
(2)$$L=\frac{Z_{2}\sin(\theta_{2})}{\omega }$$

Figure 1b shows the equivalent circuit after adding the two variable capacitors. We can deduct the following from (1) and (2):(3)$$\frac{C_{x}}{2}=\frac{\tan\left(\frac{\theta }{2}\right)}{Z\omega }$$(4)$$L_{x}=\frac{Z\sin(\theta )}{\omega }$$
where *C_x_*, *L_x_*, *θ*, and *Z* are, respectively, the capacitor, the inductance, the electrical length, and the impedance of the new adapted model.

Note that $C_{x}=2C+C_{0}$ and $L_{x}=L$; this allows for (3) and (4) to be written as follows:(5)$$C+\frac{C_{0}}{2}=\frac{\tan\left(\frac{\theta }{2}\right)}{Z\omega }$$
(6)$$L=\frac{Z\sin(\theta )}{\omega }$$

From (1) and (2) we obtain the following:(7)$$C+\frac{\tan\left(\frac{\theta_{2}}{2}\right)}{z_{2}\omega }=\frac{\tan\left(\frac{\theta }{2}\right)}{Z\omega }$$
(8)$$\frac{Z_{2}\sin(\theta_{2})}{\omega }=\frac{Z\sin(\theta )}{\omega }$$

The following equations can be deducted from (7) and (8):(9)$$\theta =2\tan^{-1}\left[Z\omega C+\frac{Z}{Z_{2}}\tan(\frac{\theta_{2}}{2})\right]$$
(10)$$Z=\frac{Z_{2}\sin(\theta_{2})}{\sin(\theta )}$$

As a recall, here, *C* is the equivalent variable capacity of the used beam-lead varactor (210 × 610 μm) purchased from the M/A-COM Company [24].

### 2.2. Even and Odd Mode Analysis of the Proposed Coupler

The proposed coupler, as illustrated in Figure 2b, is made of four branches. Two branches are characterized by an impedance of Z_1_ and an electrical length of *θ*_1_ = 90°, and the other two have an impedance of Z and an electrical length of *θ*_2_, and two variable capacitors *C*_1_ and *C*_2_, which are electronically controlled by the DC voltage applied over the varactor diode.

Each port is fed with a transmission line of impedance *Z*_0_. The input port is chosen as port 1, while ports 2 and 4 are for the output, with port 3 remaining isolated. The symmetrical nature of the coupler allows to be analyzed in two horizontal segments using even–odd mode analysis to obtain the coupler parameters. The electrical lengths of the two vertical branches will be noted as *θ* and *θ*′, where *θ* represents the branch with *C*_1_ and *θ*′ is the branch with *C*_2_. 

As a recall, the design aims to achieve a variable phase difference, *ψ*, while maintaining a coupling coefficient of −3 ± 0.5 dB. To obtain solutions more easily, we can reuse all of the results obtained after the even–odd mode analysis in [17,18], precisely the closed form equations:(11)$$P=\left|\frac{S_{21}}{S_{41}}\right|$$
(12)$$\psi =∠S_{41}-∠S_{21}$$

P and ψ are, respectively, the power division ratio and the phase difference between the output ports 2 and 4.
(13)$$Z_{1}=Z_{0}P\left|sin\psi \right|$$
(14)$$Z_{2}=\frac{Z_{0}P\left|sin\psi \right|}{\sqrt{1+P^{2}{sin}^{2}\psi }}$$
where *Z*_2_ represents the impedance before adding capacitors.
(15)$$\theta_{1}=\frac{\pi }{2}$$
(16)$$\theta +\theta^{\prime }=\pi$$
(17)$$\theta ={tan}^{-1}\left(\frac{Z_{0}tan\psi }{Z_{1}}\right)$$
(18)$$\theta^{\prime }=\pi -{tan}^{-1}\left(\frac{Z_{0}tan\psi }{Z_{1}}\right)$$

By utilizing a series of Equations (11)–(18), the fundamental design of a coupler operating at 3.5 GHz was established. Equations (13) and (14) show that the coupling coefficient depends on the output phase difference, ψ, and the impedances, *Z*_0_, *Z*_1_, and *Z*_2_. To have a tolerable coupling coefficient of −3 ± 0.5 dB at the center frequency, the output phase difference, *ψ*, must be limited to 90° ± 38°, with *Z*_1_ = *Z*_0_ = 50 Ω and *Z*_2_ = 35.35 Ω. Since *Z*_1_ = *Z*_0_, Equation (17) becomes the following:(19)ψ=θ

At 3.5 GHz, after adding the capacitors, *θ*_2_ and *Z*_2_ are changed to *θ* and *Z*, and their values are 79° and 46 Ω, respectively. These values have been optimized to retain a good performance of a conventional 3-dB/90° hybrid coupler, in terms of good matching, good isolation, and good power ratio, while the phase shift is now *ψ* (see Equation (20)), instead of 90°. This optimization was validated by the CST simulation software.

From (9) and (19), it can be deduced as follows:(20)$$\psi =2\tan^{-1}\left[Z\omega c_{1}+\frac{Z}{Z_{2}}\tan(\frac{\theta_{2}}{2})\right]$$

It is clear from Equation (20) that the coupler’s output phase difference depends mainly on *C*_1_, since the other parameters are constants. On the other hand, C_2_ capacitors will depend on *C*_1_ and it is useful to have the necessary *θ*’ to maintain the condition in (18). 

The two variable capacitors are independently controlled by two biasing voltages: *V*_1_ for *C*_1_ and *V*_2_ for *C*_2_.

Table 1 contains a list of the optimized parameter values, and Figure 2b provides the definitions of each parameter.

## 3. Simulation and Measurement Results

The proposed coupler has been designed and optimized using Computer Simulation Technology (CST STUDIO SUITE 2018) software. For fabrication, Rogers RT/Duroid 5880 with εr = 2.2, h = 0.5 mm is used as the substrate, and for the varactor diodes, four Beam Lead MA46580-1209 [24] are used. This varactor diode offers a capacitance range from 0.165 pF to 1.23 pF, at 3.5 GHz, by adjusting the DC voltage from 0 to 19 V. A photograph of the fabricated coupler is shown in Figure 3. The input port is chosen as port 1, while ports 2 and 4 are for the output, with port 3 remaining isolated.

Only three values of output phase difference will be processed and analyzed. The two extreme values, 52° and 128°, and the value of 90° allowing to have the conventional structure. We will note these three choices as follows:Config_1: This configuration represents the minimum value of the phase difference, with *ψ* = 52°.Config_2: This configuration represents the conventional case, with ψ = 90°.Config_3: This configuration represents the maximum value of the phase difference, with *ψ* = 128°.

Table 2 shows in detail the required voltage and capacitance values for the three configurations.

Figure 4a shows the comparison of simulated and measured results of output phase shift for the three chosen configurations. It can be noted that the measured results show a good correspondence with simulations. Contrarily to what was expected from simulations as a maximum phase difference ranging from 52° to 128°, the measured maximum one is from 53° to 120° occurs at 3.8 GHz instead of 3.5 GHz (not shown in Figure 4a because our target frequency range is 3.4 GHz to 3.6 GHz). This means, there is a phase mismatch of less than 8° which is more observed for Config_3 and a frequency shift of 300 MHz. This can be due to the used bonding method used for mounting the varactor diodes and mainly to the misconnection (i.e., weld the connectors) of the SMA connector at the different ports that were not included in the simulation.

Therefore, it can be seen that the maximum simulated range occurs at bias points of (V_1_ = 2.3 V, V_2_ = 17 V) and (V_1_ = 17 V, V_2_ = 2.3 V), respectively, for 52° and 128°, while the measured range is obtained at bias points of (V_1_ = 1 V, V_2_ = 18 V) and (V_1_ = 18 V, V_2_ = 1 V), respectively, for 53° and 117° (but the maximum value is about 120° measured at 3.8 GHz). This difference in DC voltage to obtain the suitable phase difference can probably be caused by the effects of some parasitic elements that are not captured in the whole equivalent developed model of this varactor in [25], since the supplier does not provide any model. Figure 4b–d shows the comparison of the simulated results with those measured by the S-parameters. As can be observed, a good reflection and isolation of less than −10 dB over the entire desired band, 3.4 GHz to 3.6 GHz, for all three configurations, is achieved. For the insertion loss, additional losses of about 1~2 dB are observed. This is due to the unavoidable losses caused by the varactor diodes. The insertion losses increase as the phase difference moves further away from 90°, either greater or less than 90°. Therefore, the insertion losses reach their maximum in Config1 (53°) and Config3 (120°). This implies that, regardless of the configuration chosen, the insertion losses will always be lower than those observed in Config1 and Config3.

In Figure 5, the phase difference obtained versus DC bias at 3.5 GHz is illustrated for both simulations and measurements. Therefore, as can be observed, a continuously variable phase-shift range of 76° is achieved in simulations, while the measured one is about 67°. This measured phase shift range versus DC bias exhibits the same general behavior as the simulated one. Table 3 shows in detail the required voltage for biasing DC points.

To emphasize the advantages of the proposed structure, Table 4 provides a comparison between this work and previous studies. In comparing the proposed coupler with other designs, the proposed coupler offers continuous tunability of the output phase, while minimizing additional insertion losses. It also maintains satisfactory isolation and reflection coefficients, all with a standard and simple structure.

## 4. Conclusions

This paper presents the design of a coupler with tunable output phase differences using varactor diodes. The proposed structure is simple, easy to manufacture, and above all, low cost. The proposed coupler has been manufactured, tested, and validated. A good agreement was obtained between the measured and simulated results. This coupler could be used in a Butler matrix, an option that will improve the performance of the matrix by offering a variable output phase difference.

## Figures and Tables

**Figure 1 micromachines-15-00838-f001:**
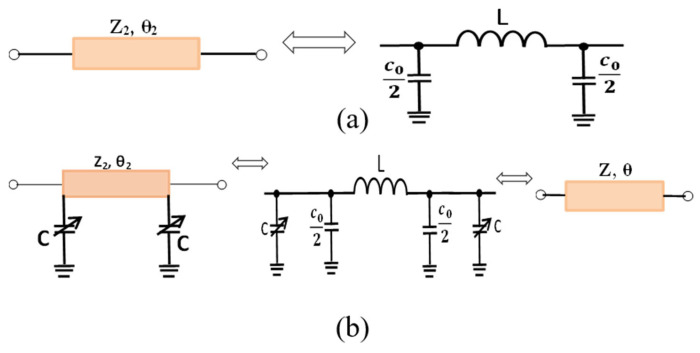
(**a**) Equivalent Pi network circuit of transmission line [23] and (**b**) represent equivalent circuit after added two variable capacitors.

**Figure 2 micromachines-15-00838-f002:**
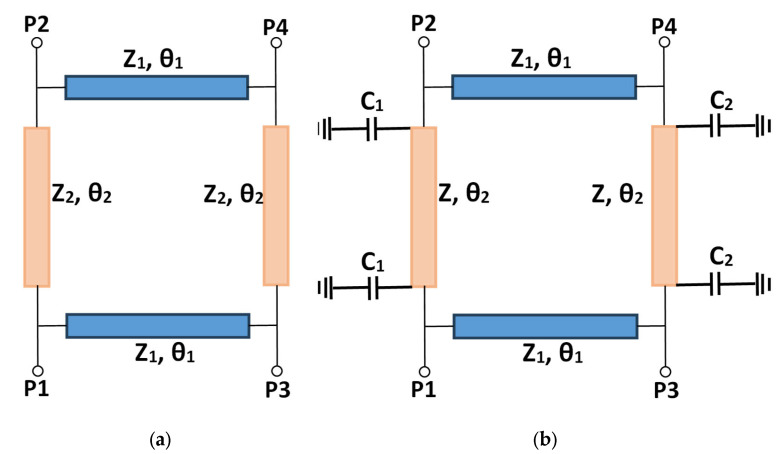
Schematic representation of (**a**) conventionnel coupler and (**b**) the proposed coupler.

**Figure 3 micromachines-15-00838-f003:**
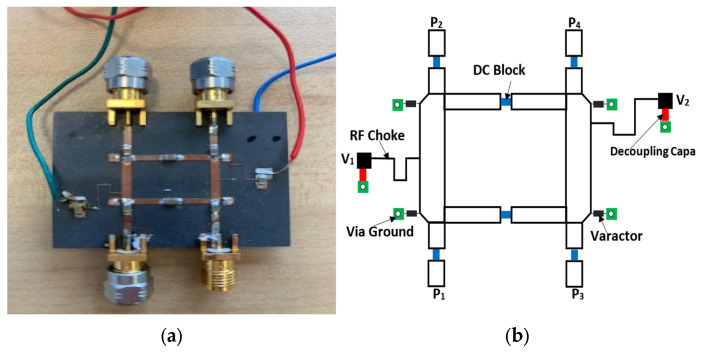
(**a**) Photograph of the fabricated coupler and (**b**) the schematic equivalent with related element information.

**Figure 4 micromachines-15-00838-f004:**
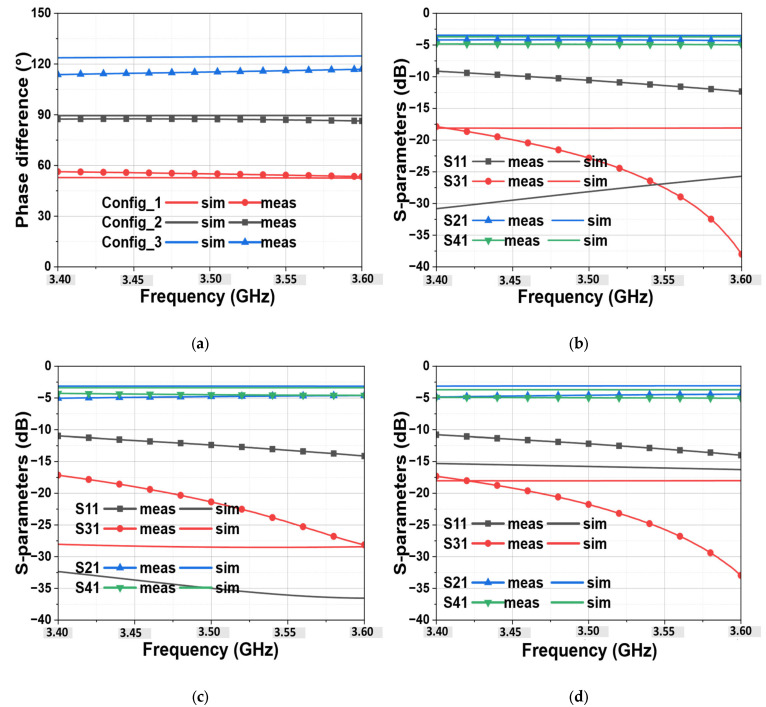
Simulated and measured results: (**a**) the output phase difference vs. frequency of the three configurations, (**b**–**d**) represent respectively the S-parameters of Config_1, Config_2, and Config_3.

**Figure 5 micromachines-15-00838-f005:**
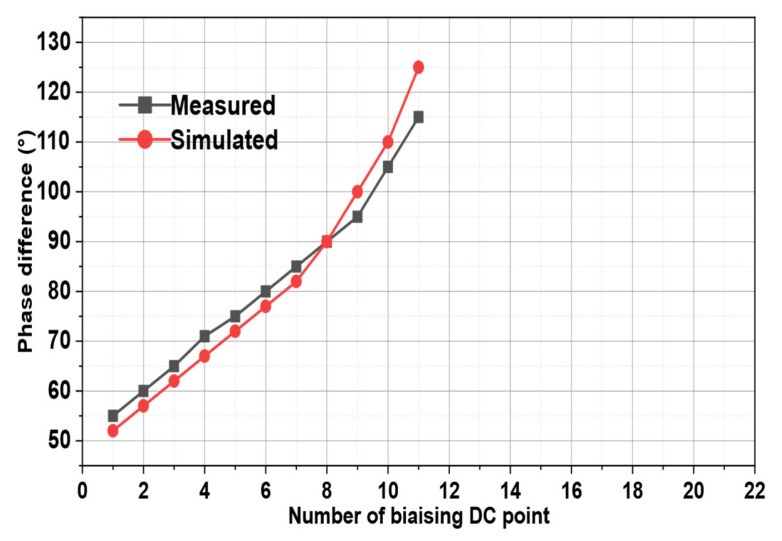
The output phase difference versus DC Bias at 3.5 GHz.

**Table 1 micromachines-15-00838-t001:** Optimized parameters of the proposed coupler.

Parameters	Initial Values	Optimized Values after Adding the Capacitors
θ_1_	90°	90°
θ_2_	90°	-
θ	-	79°
Z_0_	50 Ω	50 Ω
Z_1_	50 Ω	50 Ω
Z_2_	35.35 Ω	-
Z	-	46 Ω

**Table 2 micromachines-15-00838-t002:** Required voltage and capacitance for the designed coupler.

	Simulated		Measured	
	Biasing Voltages (V)	Capacitance (pF)	Biasing Voltages (V)	Capacitance (pF)
Config_1	V_1_ = 2.3 V_2_ = 17	C_1_ = 0.8 C_2_ = 0.1	V_1_ = 1 V_2_ = 18	C_1_ = 1.13 C_2_ = 0.09
Config_2	V_1_ = V_2_ = 4.2	C_1_ = C_2_ = 0.53	V_1_ = V_2_ = 4.2	C_1_ = C_2_ = 0.53
Config_3	V_1_ = 17 V_2_ = 2.3	C_1_ = 0.1 C_2_ = 0.8	V_1_ = 18 V_2_ = 1	C_1_ = 0.09 C_2_= 1.13

**Table 3 micromachines-15-00838-t003:** Required voltage for biasing DC points.

Biasing DC Points	Biasing Voltages (V)	
	Simulated	Measured
1	V_1_ = 2.3; V_2_ = 17	V_1_ = 1; V_2_ = 18
2	V_1_ = 2.5; V_2_ = 13	V_1_ = 1.5; V_2_ = 16
3	V_1_ = 2.7; V_2_ = 9	V_1_ = 2; V_2_ = 14
4	V_1_ = 2.9; V_2_ = 8	V_1_ = 2.3; V_2_ = 12
5	V_1_ = 3.1; V_2_ = 7	V_1_ = 2.7; V_2_ = 9
6	V_1_ = 3.5; V_2_ = 6	V_1_ = 3.3; V_2_ = 8
7	V_1_ = 3.9; V_2_ = 5	V_1_ = 3.8; V_2_ = 6
8	V_1_ = 4.2; V_2_ = 4.2	V_1_ = 4.2; V_2_ = 4.2
9	V_1_ = 4.6; V_2_ = 3.7	V_1_ = 6; V_2_ = 3.8
10	V_1_ = 7.5; V_2_ = 3	V_1_ = 9; V_2_ = 2.7
11	V_1_ = 17; V_2_ = 2.3	V_1_ = 18; V_2_ = 1

**Table 4 micromachines-15-00838-t004:** Comparison between the proposed coupler and other recent works.

	fc (GHz)	Return Loss (dB)	Isolation (dB)	Continuous Tuning	Phase Difference Range	Additional Insertion Loss (dB)
[15]	2.5	10	10	No	45°, 60°	<2
[18]	2.4	15	14	No	60°, 90°, 120°	<2
[19]	1.8	12	14	Yes	60°~120°	<1.4
[20]	1	8	12	Yes	45°~135°	<3
This work	3.5	10	17	Yes	53°~120°	<2

## Data Availability

Data are contained within the article.

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
