# Peer review of "Hybrid Coupler Used as Tunable Phase Shifter Based on Varactor Diodes"

_micromachines, 2024, doi:10.3390/mi15070838_

Round 1
Reviewer 1 Report
Comments and Suggestions for Authors
A electrical tuned phase shifter is definitely interest and it is important in numbers of application. In this work, an effective method is propsoed by using the varactors for the tunable transmission line, however, it doesn't show sufficient novelity of the proposed structure. Also, it is sure with capacitive loaded on the TLs (Z2, theta2) section, the overall hybrid performance will be distorated, but it hasn't demostrated any compensated approach applied to the TLs section (Z1, theta2).
Author Response
Dear Reviewer 1,
I hope this letter finds you well. I would like to express my sincere gratitude for taking the time to review my manuscript. Your feedback and constructive comments were immensely valuable and have significantly contributed to enhancing the quality of the manuscript.
Based on your invaluable feedback, I have made several revisions to the manuscript. Not only have I addressed the specific comments and suggestions you provided, but I also took the opportunity to make additional clarifications. In order to facilitate an easy review, all the modifications, corrections, and additions to the manuscript have been written in red.
I hope that you will find the revised manuscript to be improved. I am optimistic that the modifications will address the concerns raised, and I am hopeful that the revised version of my manuscript is found to be suitable for publication.
Thank you very much for your time and consideration.
Sincerely,
Below are the responses to the comments:
Comment 1:
A electrical tuned phase shifter is definitely interest and it is important in numbers of application. In this work, an effective method is propsoed by using the varactors for the tunable transmission line, however, it doesn't show sufficient novelity of the proposed structure. Also, it is sure with capacitive loaded on the TLs (Z2, theta2) section, the overall hybrid performance will be distorated, but it hasn't demostrated any compensated approach applied to the TLs section (Z1, theta2).
Response 1:
Thank you for your comment.
We agree that the addition of capacitors can distort the overall performance of the coupler. To address this, we modified the impedance Z2 and the electrical length Theta2, which initially had values of Z2=35.35Ω and Theta2=90°. To maintain the coupler's performance after adding the capacitors, we optimized Z2 and Theta2 to obtain new values of Z2=Z=46Ω and Theta2=Theta=79°. This optimization allowed us to preserve the coupler's performance in terms of good matching, good isolation, and a good power ratio, as shown in Figure 4. This explanation is included in the paragraph (lines 176 to 180).
Reviewer 2 Report
Comments and Suggestions for Authors
This paper propose a hybrid coupler with a continuously tunable phase difference based on varactor 2 diodes, from this perspective, the paper is interesting. The following issues should be addressed before the paper can be considered.
1. The authors state “a tunable output phase difference from 52 to 128” is achieved, what is possible applications of this phase difference?
2. The innovation should be enhanced in introduction part.
3. As to coupler, the following work using composite Right/Left-handed technique may help to setup a solid review in this area. Broadband balun using fully artificial fractal-shaped composite right/left handed transmission line; Compact Butler matrix using composite right/left handed transmission line; Novel composite right/left handed transmission lines using fractal geometry and compact microwave devices application; Also, a review article: Microstrip rat-race couplers.
4. The difference between measured and simulated S31 seems a little large, what is the reason?
5. Also as to S21 and S41, they are a little large than theoretical one 3dB, what is the reason?
6. It is seems that the difference in phase difference of Config_1 is larger than that in cases of Config_2, 207 and Config_3, why? They should discuss in depth.
7. In table 2, Why there is big deviation of used Biasing Voltages in Simulated and Measured cases in Config_1 and Config_3?
The authors state that “This coupler could be used in a Butler matrix”, does the variable output phase difference affect the output 3dB amplitude?
Comments on the Quality of English LanguageMinor editing of English language required
Author Response
Dear Reviewer 2,
I hope this letter finds you well. I would like to express my sincere gratitude for taking the time to review my manuscript. Your feedback and constructive comments were immensely valuable and have significantly contributed to enhancing the quality of the manuscript.
Based on your invaluable feedback, I have made several revisions to the manuscript. Not only have I addressed the specific comments and suggestions you provided, but I also took the opportunity to make additional clarifications. In order to facilitate an easy review, all the modifications, corrections, and additions to the manuscript have been written in red.
I hope that you will find the revised manuscript to be improved. I am optimistic that the modifications will address the concerns raised, and I am hopeful that the revised version of my manuscript is found to be suitable for publication.
Thank you very much for your time and consideration.
Sincerely,
Below are the responses to the comments:
Comments 1: The authors state “a tunable output phase difference from 52 to 128” is achieved, what is possible applications of this phase difference?
Response 1:
Thank you for your question.
This tunable phase difference can be very beneficial when applied in a Butler matrix. A Butler matrix generates four beams, and with the proposed coupler, each beam can be steered within a range of 12°, allowing coverage of all areas between the beams. For more details on the Butler matrix, please refer to our publication DOI: 10.1109/ISAP57493.2023.10388536.
Comments 2: The innovation should be enhanced in introduction part.
Response 2:
Thank you for your suggestion. The introduction has been improved.
Comments 3: As to coupler, the following work using composite Right/Left-handed technique may help to setup a solid review in this area. Broadband balun using fully artificial fractal-shaped composite right/left handed transmission line; Compact Butler matrix using composite right/left handed transmission line; Novel composite right/left handed transmission lines using fractal geometry and compact microwave devices application; Also, a review article: Microstrip rat-race couplers.
Response 3:
Thank you for your suggestion. Some References to the suggested works using composite Right/Left-handed techniques have been included to enhance the literature review. (line 33 to 34 and line 48 to 50).
Comments 4: The difference between measured and simulated S31 seems a little large, what is the reason?
Response 4:
Thank you for your question.
The S31 is the isolation coefficient and for a good isolation this coefficient has to be less than -10dB. Compared to the simulation, the measured S31 is getting more improved while increasing frequency. The difference between measured and simulated S31 results can be due to the misconnection of the SMA connector to the various ports. In simulations, only a CST lumped port was used instead of a SMA connector, this means this connection that the SMA connection was not taken into consideration.
Comments 5: Also, as to S21 and S41, they are a little large than theoretical one 3dB, what is the reason?
Response 5:
Thank you for your question.
The additional losses of approximately 1 dB ∼ 2 dB can be attributed to several factors. These include the transmission line losses, the inherent losses introduced by the varactor diodes, and the losses caused by the SMA connectors.
Comments 6: It is seems that the difference in phase difference of Config_1 is larger than that in cases of Config_2, and Config_3, why? They should discuss in depth.
Response 6:
We believe you are referring to Config_3 rather than Config_1, as Config_1 and Config_2 show good agreement with the simulations. Indeed, there is a phase shift discrepancy between the simulations and measurements for Config_3. This may be due to the bonding method used to mount the varactor diodes and, more importantly, to the poor connection (i.e., soldering the connectors) of the SMA connector to the various ports, which were not included in the simulation.
This is explained on line 224 to 231.
Comments 7: In table 2, Why there is big deviation of used Biasing Voltages in Simulated and Measured cases in Config_1 and Config_3?
Response 7:
Thank you for your question.
This difference in DC voltage to get the suitable phase difference can be probably caused by the effects of some parasitic elements that is not captured in the whole equivalent in-house developed model of this varactor in [25] since the suppler doesn’t provide any model. (See lines 235 to 238).
Comments 8: The authors state that “This coupler could be used in a Butler matrix”, does the variable output phase difference affect the output 3dB amplitude?
Response 8:
Thank you for your question.
Certainly, the variance in the output phase will affect the output amplitude. However, as you know, the Butler matrix is essentially based on couplers. To analyze this effect, we can look at the S21 and S41 results obtained from the proposed coupler. According to these results, it appears that the effect of the output phase variance on the amplitude is acceptable.
Reviewer 3 Report
Comments and Suggestions for Authors
In this manuscript, the author proposes a hybrid coupler with a continuously variable output phase difference, analyzes the proposed coupler's design method, and completes the experimental verification. The demonstration of the concept is comprehensive, including analysis and experimental results. The manuscript presents an innovative design of a hybrid coupler with a continuously tunable output phase difference based on varactor diodes. Still, it needs further refinement in certain technical details and presentation of results.
(1) The port name and circuit element-related information could be marked in Figure 3, and a higher-resolution picture could be used.
(2) In Section 2.1, it showns that Cx equals 2C+C0 by referring to Figure 2(a) and equations (1) and (3).
(3) Figure 4(a) shows the phase difference measured from which two ports should be accurately stated. The legend information illustrated in Figure 5 could be presented in tabular form.
(4) Under the three voltage configurations in Table 2, the coupler's insertion loss and return loss perform well. Does the coupler still meet good insertion and return losses under other configurations, as shown in Figure 5?
(5) A comparison table with similar works should be provided to show the proposed design's advantages and disadvantages.
(6) Why 3.5GHz was chosen as the operating frequency for the design?
(7) Why did you choose this MA46580-1209 diode? How are the component parameters (e.g., capacitance value of the varactor diode, bias voltage, etc.) used in the simulation set?
(8) The manuscript mentions that the effects of some parasitic elements may not be fully accounted for in the simulation because the supplier did not provide a complete equivalent model of the varactor diode. Does this have a significant impact on the accuracy of the simulation results? How were the models used in the simulations considered?
(9) The manuscript proposes that the actual operating frequency point for the maximum phase difference range is shifted to 3.8 GHz, but no relevant experimental results are presented. If you think this may be caused by the soldering and connection of the device, can you explain further?
(10) The manuscript mentions that previous studies have explored multiple ways to tune the coupler's performance. A more detailed comparative analysis should be added to highlight the proposed design's advantages and potential room for improvement.
Comments on the Quality of English LanguageIt is OK for me.
Author Response
Dear Reviewer 3,
I hope this letter finds you well. I would like to express my sincere gratitude for taking the time to review my manuscript. Your feedback and constructive comments were immensely valuable and have significantly contributed to enhancing the quality of the manuscript.
Based on your invaluable feedback, I have made several revisions to the manuscript. Not only have I addressed the specific comments and suggestions you provided, but I also took the opportunity to make additional clarifications. In order to facilitate an easy review, all the modifications, corrections, and additions to the manuscript have been written in red.
I hope that you will find the revised manuscript to be improved. I am optimistic that the modifications will address the concerns raised, and I am hopeful that the revised version of my manuscript is found to be suitable for publication.
Thank you very much for your time and consideration.
Sincerely,
Below are the responses to the comments:
Comments 1: The port name and circuit element-related information could be marked in Figure 3, and a higher-resolution picture could be used.
Response 1: Thank you for your suggestion. A schematic equivalent with related element information has been added to Figure 3.
Comments 2: In Section 2.1, it shows that Cx equals 2C+C0 by referring to Figure 2(a) and equations (1) and (3).
Response 2:
Thank you for your remark. you're right, Cx = 2C+C0 and not C+C0/2
The mistake has been corrected.
Comments 3: Figure 4(a) shows the phase difference measured from which two ports should be accurately stated. The legend information illustrated in Figure 5 could be presented in tabular form.
Response 3: Thank you for your suggestion. A paragraph explaining the input port, the two output ports and the isolation port chosen during measurements has been added (line 204 to 205). The information in the legend shown in figure 5 has been presented in tabular form (see table 3).
Comments 4: Under the three voltage configurations in Table 2, the coupler's insertion loss and return loss perform well. Does the coupler still meet good insertion and return losses under other configurations, as shown in Figure 5?
Response 4: Thank you for your question.
The insertion losses increase as the phase difference moves further away from 90°, either greater or less than 90°. Therefore, the insertion losses reach their maximum in Config1 (52°) and Config3 (128°). This implies that, regardless of the configuration chosen, the insertion losses will always be lower than those observed in Config1 and Config3.
The above paragraph has been added to the manuscript. (line 243 to 247)
Comments 5: A comparison table with similar works should be provided to show the proposed design's advantages and disadvantages.
Response 5: Thank you for your suggestion. A comparison of the proposed coupler and other recent work has been added (table 4).
Comments 6: Why 3.5GHz was chosen as the operating frequency for the design?
Response 6: Thank you for your question.
The choice of 3.5 GHz was made to target 5G applications. We believe that this work could meet the requirements of 5G, especially in terms of beam steering.
Comments 7: Why did you choose this MA46580-1209 diode? How are the component parameters (e.g., capacitance value of the varactor diode, bias voltage, etc.) used in the simulation set?
Response 7: Thank you for your question.
The MA46580-1209 varactor diode was chosen because it offers a capacitance tuning range that meets our requirements and is readily available on the market. The capacitance range of the selected varactor diode is between 0.165 pF and 1.23 pF at 3.5 GHz, by adjusting the DC bias from 0 to 19 V. (this paragraph has been added to the manuscript lines 202 to 204).
To define the component values of the varactor diode during simulations, we based our work on a model of this diode developed in [25], which provides the capacitance values corresponding to the applied voltages.
Comments 8: The manuscript mentions that the effects of some parasitic elements may not be fully accounted for in the simulation because the supplier did not provide a complete equivalent model of the varactor diode. Does this have a significant impact on the accuracy of the simulation results? How were the models used in the simulations considered?
Response 8: Thank you for your question. The impact is not on the results themselves but rather on the accuracy of the values assigned to the components during the simulation to obtain these results.
How were the models used in the simulations considered? à CST simulator has the ability to combine the EM and schematic (here, varactor model) simulation which called co-simulation.
Comments 9: The manuscript proposes that the actual operating frequency point for the maximum phase difference range is shifted to 3.8 GHz, but no relevant experimental results are presented. If you think this may be caused by the soldering and connection of the device, can you explain further?
Response 9: The results at 3.8 GHz are not presented because our target frequency range is between 3.4 GHz and 3.6 GHz, and the results for other configurations are satisfactory within this range.
It is common for measured results to sometimes deviate from simulated results, which can be attributed to factors not taken into account in the simulations, such as the gluing method used to mount the varactor diodes or poor connection (i.e. soldering of connectors) of the SMA connector to the various ports.
Comments 10: The manuscript mentions that previous studies have explored multiple ways to tune the coupler's performance. A more detailed comparative analysis should be added to highlight the proposed design's advantages and potential room for improvement.
Response 10: Thank you for your suggestion. We have improved the introduction.
Round 2
Reviewer 1 Report
Comments and Suggestions for Authors
I have no further comments and questions.